# Soybean Calmodulin-Binding Transcription Activators, GmCAMTA2 and GmCAMTA8, Coordinate the Circadian Regulation of Developmental Processes and Drought Stress Responses

**DOI:** 10.3390/ijms241411477

**Published:** 2023-07-14

**Authors:** Dongwon Baek, Hyun Min Cho, Ye Jin Cha, Byung Jun Jin, Su Hyeon Lee, Mi Suk Park, Hyun Jin Chun, Min Chul Kim

**Affiliations:** 1Plant Molecular Biology and Biotechnology Research Center, Gyeongsang National University, Jinju 52828, Republic of Korea; dw100@hanmail.net (D.B.); misugip@hanmail.net (M.S.P.); 2Division of Applied Life Science (BK21 Four), Gyeongsang National University, Jinju 52828, Republic of Korea; hmcho86@gnu.ac.kr (H.M.C.); cdw3280@naver.com (Y.J.C.); leesuhyeon86@gmail.com (S.H.L.); 3Institute of Agriculture and Life Science, Gyeongsang National University, Jinju 52828, Republic of Korea; scv5789@naver.com

**Keywords:** GmCAMTA, circadian rhythm, development, drought stress

## Abstract

The calmodulin-binding transcription activators (CAMTAs) mediate transcriptional regulation of development, growth, and responses to various environmental stresses in plants. To understand the biological roles of soybean *CAMTA* (*GmCAMTA*) family members in response to abiotic stresses, we characterized expression patterns of 15 *GmCAMTA* genes in response to various abiotic stresses. The *GmCAMTA* genes exhibited distinct circadian regulation expression patterns and were differently expressed in response to salt, drought, and cold stresses. Interestingly, the expression levels of *GmCAMTA2*, *GmCAMTA8*, and *GmCAMTA12* were higher in stem tissue than in other soybean tissues. To determine the roles of *GmCAMTAs* in the regulation of developmental processes and stress responses, we isolated *GmCAMTA2* and *GmCAMTA8* cDNAs from soybean and generated Arabidopsis overexpressing transgenic plants. The *GmCAMTA2*-OX and *GmCAMTA8*-OX plants showed hypersensitivity to drought stress. The water in the leaves of *GmCAMTA2*-OX and *GmCAMTA8*-OX plants was lost faster than that in wild-type (WT) plants under drought-stress conditions. In addition, stress-responsive genes were down-regulated in the *GmCAMTA2*-OX and *GmCAMTA8*-OX plants under drought stress conditions compared to WT plants. Our results suggest that *GmCAMTA2* and *GmCAMTA8* genes are regulated by circadian rhythms and function as negative regulators in development and drought stress responses.

## 1. Introduction

The Ca^2+^-dependent signal cascades in plants physiologically regulate the division and elongation of cells, stomatal movement, and developmental processes in plant tolerance or adaptation responses to abiotic stresses [1,2]. These signal cascades included calmodulin (CaM) and calmodulin-like proteins (CMLs), calcineurin B-like proteins (CBLs), and calcium-dependent protein kinases (CDPKs), playing a role as Ca^2+^ sensors [1]. The CaMs with one or more EF-hand Ca^2+^-binding motifs are associated with a multitude of biochemical reactions, such as the control of transcriptional expression and enzyme activity [2,3]. Although the Ca^2+^ ion lacks cellular levels to activate CaM under natural conditions, when exposed to abiotic stresses, the Ca^2+^ levels are rapidly increased, generating the Ca^2+^ signals, and there is a boost in the activity of CaM [4]. CaM is the best characterized of the Ca^2+^ receptors, and the Ca^2+^–CaM complex regulates the ion channels, transporters, transcription factors (TFs), and various enzymes, including protein kinase, phosphatases, and metabolic enzymes [5,6]. The CaM-driven TFs, such as calmodulin-binding transcription activators (CAMTAs), DREB, MYB, WRKY, NAC, bZIP, bHLH, and MADS-box play a pivotal role in multiple cellular processes, including plant stress responses and developmental processes [1,6].

The expression patterns of *CAMTAs* show significant differences in the extensive developmental stages and plant stress responses [1,6]. In *Arabidopsis*, the expression of *AtCAMTA6* (*AT3G16940*) shows a high level in mature leaves and siliques compared with other tissues [6]. In addition, the expression levels of *AtCAMTA2* (*AT5G64220*) in seeds and young leaves are significantly lower than those in other tissues [6]. In tomato (*Solanum lycopersicum*), *PtCAMTA4* (*JN566050*) shows high levels of transcripts in turning and orange fruit stages, while *PtCAMTA2* (*JN566047*) is highly expressed in all fruit developmental stages except the mature green and breaker stages [7]. In the tea (*Camellia sinensis*) plant, the expression of *CsCAMTA1* (*TEA025813.1*) is at higher levels in old leaves than those in other tissues [8]. In black cottonwood (*Populus trichocarpa*), *PtCAMTA1* (*POPTR_0001s13700*) is highly expressed in the mature leaves, while *PtCAMTA2* (*POPTR_0005s07660*) and *PtCAMTA3* (*POPTR_0007s05410*) are weakly expressed in the roots compared to other tissues [9]. The expression levels of all seven rice *OsCAMTA* genes are displayed more in leaves after flowering compared with before flowering [6]. The *MuCAMTA1* (*Musa acuminate*; *LOC1039776*) and *phavuCAMTA1* (*Phaseolus vulgaris*; *PHAVU_006G206400g*) genes are highly expressed under drought stress conditions [10,11]. In the root of maize (*Zea mays*), the expression of *ZmCAMTA1* (*GRMZM2G171600*) is significantly decreased to cold stress, but increased in response to salt stress and jasmonic acid (JA) treatment [12]. In *Medicago truncatula*, the expression of *MtCAMTA7* (*Medtr8g090205*) is induced by ABA treatment [13]. Therefore, the expression of *CAMTA* genes presents substantial specificity in the developmental stages and tissues, as well as in stress responses.

CAMTAs in plants are known to be positive or negative regulators in plant development processes, as well as in biotic and abiotic stress responses [14,15,16]. *AtCAMTA1* (*AT5G09410*) positively regulates the ABA-dependent response for enhancing plant tolerance to drought stress [14]. The *atcamta1* mutants show hypersensitivity to drought stress by regulating stress-responsive genes, such as *RD26*, *ERD7*, *RAB18*, *LTPs*, and *COR78* [14]. *AtCAMTA3* (*AT2G22300*) functions as a negative regulator in plant defense responses against bacteria [17] and fungi [18]. The *atcamta3* mutants show enhanced resistance to *Pseudomonas syringae* pv. *tomato* DC3000 (*Pst*DC3000) and *Botrytis cinerea* by the high expression of *pathogenesis-related* (*PR*) genes [17,18]. In contrast, *AtCAMTA3* acts as a positive regulator in plant tolerance to freezing stress [15]. A mutation of *atcamta3* weakens the freezing tolerance caused by reducing the cold-induced accumulation of *CBF* gene transcript levels during exposure to low temperatures [15]. *AtCAMTA1* and *AtCAMTA5* (*AT4G16150*) positively regulate the pollen developmental process by increasing the expression of *Arabidopsis V-PPase 1* (*AVP1*) [19]. Transcriptome profiling using a *camta6* mutant under salt stress shows that 638 upregulated and 1242 downregulated genes were involved in *AtCAMTA6*-dependent manners [20]. In rice, *OsCBT*/*OsCAMTA5* (*LOC_Os07g30774*) plays a negative role in the plant defense response against pathogens with the high expression of *PR* genes, such as *PR1*, *PR4*, *PR10*, and *PBZ* [21]. Rice *oscbt-1* mutants show enhanced resistance against both rice blast fungus (*Magnaporthe grisea*) and bacteria (*Xanthomonas oryzae*) [21]. Soybean *GmCAMTA12* (*Glyma.17G031900*) regulates the expression of its regulatory network-related genes with the *CGCG*/*CGTTG* motif under drought stress in *Arabidopsis* and soybean hairy roots [22]. Ectopic expression of peach (*Prunus persica*) CAMTA gene, *PpCAMTA1* (*Prupe.1G108700*) in *Arabidopsis atcamta2/3* double mutants suppresses salicylic acid (SA) biosynthesis and the expression of SA-related genes in the plant defense response against *Pst*DC3000 [23]. Thus, CAMTAs in plants play crucial roles as transcription factors in both biotic and abiotic stress responses.

Drought stress leads to ABA accumulation in ABA-dependent drought stress response, such as the closing of stomata [24]. The ABA levels in plants oscillate energetically during the daytime, reaching maximum levels of ABA towards noon, then decreasing to the lowest level during the night [25]. Rhythmic expression of ABA-responsive genes, such as *β-glucosidase* (*BG1*) [25] and *9-cis-epoxycarotenoid dioxygenase 3* (*NCED3*) [26], control the stomata opening for minimizing water loss [27]. In addition, plant drought tolerance was regulated by circadian core oscillators [27]. These core oscillators included *Lov kelch protein 2* (*LKP2*) [28], *Pseudo-response regulator proteins* (*PRR5*, *7*, and *9*) [29], *Gigantea* (*GI*) [30], and *Timing of CAB 1* (*TOC1*) [31]. Mutation or overexpression of these genes shows sensitive or tolerant phenotypes when exposed to drought stress [28,29,30,31]. These studies suggest that rhythmic accumulation of ABA levels and rhythmic expression of drought-responsive genes have an effect on determining the stress-sensitive or tolerant phenotype to drought stress.

The 15 *GmCAMTAs* have been identified in soybeans by the SoyBase database (http://soybase.org/, accessed on 8 November 2022) (Appendix A), and the analysis of their gene expression shows that the expression patterns of *GmCAMTAs* were altered during abiotic stress (salt, drought, cold, and oxidative) and hormone (ABA, SA, and JA) responses [32]. Recently, it was reported that GmCAMTA12 plays an important role in the tolerance responses of *Arabidopsis* and soybeans to drought stress [22]. However, the functional characterization of other GmCAMTA members is still needed to cover their roles in soybean development processes and stress tolerance responses. In this study, we showed that soybean *GmCAMTA2* and *GmCAMTA8* function in the developmental processes and drought stress response.

## 2. Results

### 2.1. Circadian Rhythms Affected the Expression Patterns of GmCAMTAs under Long-Day Conditions

The stress-responsive genes in plants are rhythmically expressed in stress signaling, thereby producing daily cellular, metabolic, and physiological rhythms [24]. To investigate the influence on the transcripts of *GmCAMTAs* by daily circadian rhythms, the expression patterns of *GmCAMTAs* under long-day conditions (16 h light/8 h dark) were measured. Quantitative RT-PCR analyses showed that the expression of 15 *GmCAMTAs* accumulated with the onset of light and reached maximum access levels of their expression at ZT9 to ZT15 (Figure 1). After the light was turned off (ZT18 to ZT20), the expression levels of most *GmCAMTAs* significantly decreased more than approximately two-fold under light conditions (Figure 1). However, the expression levels of *GmCAMTA3*, *GmCAMTA5*, *GmCAMTA6*, *GmCAMTA7*, and *GmCAMTA13* at ZT18 and ZT20 were subtly reduced less than two-fold than between ZT9 and ZT15 (Figure 1). The second peak of GmCAMTAs was detected at the end of the night, between ZT21 and ZT24, even though it was weaker than the first peak between ZT9 to ZT15. In addition, in silico analysis using the Diurnal database tool showed the diurnal expression of *GmCAMTA2*, *GmCAMTA5*, *GmCAMTA8*, *GmCAMTA12*, and *GmCAMTA13* (Appendix A). Unfortunately, the diurnal expression changes of other *GmCAMTAs* were not available in the database. Therefore, it is suggested that the transcripts of *GmCAMTAs* are regulated according to circadian rhythms in the daily clock.

### 2.2. Various Abiotic Stresses Affected the Expression of GmCAMTAs

Previously, a few studies reported that the expression of *GmCAMTAs* is regulated by abiotic stresses and hormone treatment in soybean [22,32]. However, previous results had not considered the rhythmical changes of *GmCAMTAs* expression in abiotic stress responses. To accurately investigate the transcript changes of *GmCAMTAs* in response to abiotic stress treatment in view of the rhythmical changes, we measured the relative expression values of *GmCAMTA* genes under both non-stress and abiotic stress conditions; then, we carried out the normalization of the relative expression values measured under abiotic stress conditions, with the relative expression values at each time point measured under non-stress conditions. When soybean seedlings were exposed to salt stress, all of the *GmCAMTAs* were highly expressed at early times, and their expression levels increased gradually as time passed (Figure 2A). The expression levels of most *GmCAMTAs* in drought stress responses were significantly increased compared to those under non-stress conditions, except *GmCAMTA13* and *GmCAMTA15* (Figure 2B). Interestingly, *GmCAMTA1* and *GmCAMTA2* were highly expressed to cold stress; however, the expression levels of other *GmCAMTAs* were reduced during cold stress treatment (Figure 2C). ABA did not influence the expression of most *GmCAMTAs* in soybean, although *GmCAMTA1*, *GmCAMTA5*, *GmCAMTA6*, and *GmCAMTA13* were weakly expressed under exogenous ABA conditions (Figure 2D). Thus, the results indicate that the gene expressions of *GmCAMTAs* were differently influenced by various abiotic stresses.

### 2.3. The Transcripts of GmCAMTAs had Substantial Specificity in Developmental Processes

Plant developmental processes are associated with abiotic stress responses through impressive biochemical, physiological, and morphological changes [33]. We tested the changes in *GmCAMTA* expression levels in different tissues using qRT-PCR analysis. Many *GmCAMTAs* were expressed in the roots; however, *GmCAMTA2*, *GmCAMTA7*, *GmCAMTA9*, *GmCAMTA14*, and *GmCAMTA15* were weakly expressed (Figure 3, Appendix A). In the leaves, the expression of most *GmCAMTAs* was highly induced, except for those of *GmCAMTA2*, *GmCAMTA8*, and *GmCAMTA12* (Figure 3, Appendix A). Interestingly, *GmCAMTA2*, *GmCAMTA8*, and *GmCAMTA12* were relatively highly expressed in the stem tissues as compared to other *GmCAMTAs* (Figure 3, Appendix A). Unlike the expression patterns of *GmCAMTAs* in roots, leaves, and stem tissues, the expression levels of *GmCAMTAs* were similar in the flower (Figure 3, Appendix A). Our results indicated that *GmCAMTAs* were selectively expressed in each of the plant tissues, thereby showing substantial specificity in the developmental stages.

### 2.4. Overexpression of GmCAMTA2 and GmCAMTA8 Reduced Drought Tolerance in Arabidopsis

*GmCAMTA2*, *GmCAMTA8*, and *GmCAMTA12* were highly expressed in the stem (Figure 3). The overexpression of *GmCAMTA12* in *Arabidopsis* and soybean enhance plant tolerance to drought stress [22]. To investigate the functions of *GmCAMTA2* and *GmCAMTA8* in drought stress responses, we generated *Arabidopsis* transgenic plants with overexpressed *GmCAMTA2* and *GmCAMTA8* under the CAMV *35S* promoter. To establish the expression levels of *GmCAMTA2* and *GmCAMTA8* in *Arabidopsis* transgenic plants, we performed an RT-PCR analysis. According to the results of RT-PCR, we selected three individual lines, each with different expression levels of *GmCAMTA2* (*GmCAMTA2*-OX #1, #4, and #8 lines; Appendix A) and *GmCAMTA8* (*GmCAMTA8*-OX #5, #8, and #9 lines; Appendix A). We examined the tolerance of *Arabidopsis* transgenic plants overexpressing *GmCAMTA2* and *GmCAMTA8* during exposure to drought stress (Figure 4). In natural conditions, *GmCAMTA2*-OX and *GmCAMTA8*-OX plants were similar in growth and development compared with WT and empty vector-overexpressing (Vector-OX) *Arabidopsis* plants (Figure 4A). *GmCAMTA2*-OX and *GmCAMTA8*-OX plants showed hypersensitivity compared with WT and Vector-OX plants under drought stress and re-watering conditions (Figure 4A). The survival rates after re-watering under drought stress conditions were lower in the *GmCAMTA2*-OX (approximately 2.78 to 13.89%) and *GmCAMTA8*-OX (approximately 2.78 to 11.11%) than the WT (approximately 72.22 to 79.17%) and Vector-OX (approximately 70.83 to 80.56%) (Figure 4B). To confirm the drought-sensitive phenotypes in the *GmCAMTA2*-OX and *GmCAMTA8*-OX plants, we measured the water loss from detached leaves during exposure to drought stress. The water loss of the *GmCAMTA2*-OX and *GmCAMTA8*-OX plants was significantly higher than those of the WT and Vector-OX plants in response to drought stress (Figure 4C). These results suggested that the overexpression of *GmCAMTA2* and *GmCAMTA8* reduced plant tolerance to drought stress, and *GmCAMTA2* and *GmCAMTA8* function as negative regulators in drought stress response.

### 2.5. Overexpression of GmCAMTA2 and GmCAMTA8 Regulated the Transcription of Stress-Responsive Genes in Drought Stress Responses

To investigate whether overexpression of *GmCAMTA2* and *GmCAMTA8* affected the expression patterns of stress-responsive genes during drought stress responses, we performed qRT-PCR analysis of various stress-responsive genes (Figure 5). The expression levels of ABA-induced genes, *AtRD29A* and *AtRD29B*, were significantly decreased in *GmCAMTA2*-OX (*AtRD29A*; approximately 0.97- to 1.67-fold, *AtRD29B*; approximately 1.18- to 1.69-fold) and *GmCAMTA8*-OX (*AtRD29A*; approximately 1.18- to 1.61-fold, *AtRD29B*; approximately 0.94- to 1.53-fold) compared with those in WT plants (*AtRD29A*; approximately 2.61-fold, *AtRD29B*; approximately 3.60-fold) at the 60 min time point during drought stress. The *Pyrroline-5-Carboxylate Synthase 2* (*P5CS2*) gene was regulated in the biosynthesis of proline, which is an antioxidant in water stress response [34]. The expression of *AtP5CS2* was weakly decreased in *GmCAMTA2*-OX (approximately 0.75- to 0.82-fold) and *GmCAMTA8*-OX (approximately 0.62- to 0.88-fold) compared with those in WT plants (approximately 1.23-fold) at the 60 min time point during drought stress. In addition, the expression of the stress-related marker gene *AtKIN1* was decreased in *GmCAMTA2*-OX (approximately 1.60- to 1.92-fold) and *GmCAMTA8*-OX (approximately 1.37- to 1.50-fold) compared with those in WT plants (approximately 2.76-fold) at the 60 min time point during drought stress. These results suggested that the overexpression of *GmCAMTA2* and *GmCAMTA8* down-regulated the expression of stress-responsive genes during drought stress in *Arabidopsis*. Thus, GmCAMTA2 and GmCAMTA8 act as negative regulators in drought stress responses.

## 3. Discussion

The characterization of various *CAMTA* genes as core transcription factors with CaM binding sites has been reported in recent years by genome-wide analysis and expression profiles [6,7,8,9,10,11,12,13,16,32]. Although 15 *GmCAMTA* genes in soybeans have been identified, the functions of *GmCAMTAs* remained almost unknown, except for *GmCAMTA12* [22]. In this study, we showed that the expression of *GmCAMTA* genes was controlled by circadian rhythms (Figure 1). Previous studies of *CAMTA* genes have not indicated the circadian regulation of *CAMTA* gene expression. Plant cellular responses, such as gene expression, development, and metabolic processes, were circadian regulated by circadian oscillators [35,36]. Circadian responses in plants were attributable to cellular oscillations in the intracellular Ca^2+^ concentration ([Ca^2+^]) in the surrounding environment, including photoperiod and light intensity [37,38]. The calmodulin-like 24 (CML24), one of the Ca^2+^-binding proteins, regulated the circadian period in Ca^2+^-dependent signaling [36]. The most important properties of CAMTAs in plants have a high correlation with CaM proteins, as a Ca^2+^ sensor, to regulate the gene expression in Ca^2+^-dependent plant responses [1]. These facts were important to explain that the circadian regulation of *GmCAMTAs* transcripts was mediated by the cellular oscillation of [Ca^2+^].

Many *CAMTAs* in crops have been directly or indirectly associated with various developmental and cellular processes, whether through autonomously regulating a major transcription factor or targeting other transcription factors [1]. The fifteen *GmCAMTAs* in soybean [32], seven *PtCAMTAs* in *Populus trichocarpa* [9], nine *ZmCAMTAs* in maize [12], seven *MtCAMTAs* in *Medicago truncatula* [13], and fifteen *VvCAMTAs* in *Vitis vinifera* [39] were endowed with substantial specificity in the developmental stages by highly tissue-specific expression. Root-specific-induced *PtCAMTA2* and *PtCAMTA3* genes were negatively regulated under cold stress in a short time [9]. In this study, we showed that *GmCAMTA2*, *GmCAMTA8*, and *GmCAMTA12* were especially highly expressed in stem tissues compared to other *GmCAMTAs* (Figure 3). The growth and development of plant tissues, such as flowers, stems, and leaves, were greatly affected by abiotic stress [40]. The tissue-specific expression pattern of GmCAMTAs is an important characteristic for investigating the functions of GmCAMTAs in diverse molecular processes against environmental stresses. Since plants are constantly exposed to environmental stimuli during their lifetime, they have developed various adaptive mechanisms between the developmental process and stress responses [41,42].

Drought stress is an important restricting factor for plant growth and developmental processes, holding both expansion and elongation down [41]. Since drought stress triggers low turgor pressure in the cell, the osmotic maintenance of cell turgor in plants is of great importance for plant growth and survival [41]. In many crops, soybeans show a highly sensitive response to drought stress through various morphological changes [42]. When exposed to drought stress, soybean shows momentous changes in most of the tissue morphology, such as a reduction of newly developed branches and trifoliate leaves, growth inhibition of newly emerging leaves, and reduction in leaf area via repression of lamina expansion [42]. Drought stress has a negative effect on the developmental processes of stems and leaves by decreasing their elongation and expansion [41]. In particular, drought stress often leads to the expression of common morphogenesis- and stress-responsive genes through molecular mechanisms, including reactive oxygen species (ROS)- and phytohormone-dependent signaling [40]. *ATP-dependent metalloprotease 4* (*AtFtsH4*) played a tissue-specific role in maintaining the stem cell activity of shoot apical meristem (SAM) stages by ROS accumulation under high-temperature conditions [43]. *MscS-like 2* (*MSL2*) and *MSL3* were mediated to cell proliferation by regulation of *WUSCHEL* (*WUS*) gene expression in osmotic stress responses [44]. The mutation of *NADPH* genes showed unstable shoot development by decreasing the stem cell population [45]. In the case of *CAMTA* genes, the relationships between development and stress responses have been studied more in fruit species than in crops. The tomato *SISR2* and *SISR3L* transcriptions were affected by ethylene signaling during fruit development and ripening [7]. The expression of peach *PpCAMTA1* was induced in cold stress, but *PpCAMTA3* was repressed with UV-B irradiation during fruit development [23]. Our results indicated that *GmCAMTA2*, *GmCAMTA8*, and *GmCAMTA12* were significantly expressed in both stem tissues and drought stress responses (Figure 2 and Figure 3). When *GmCAMTA2* and *GmCAMTA8* were overexpressed in *Arabidopsis*, *GmCAMTA2*-OX, and *GmCAMTA8*-OX displayed the sensitive phenotype under drought stress conditions by losing water faster than WT plant (Figure 4). These results suggest that *GmCAMTA2* and *GmCAMTA8* act as negative regulators in drought stress responses. Considering the tolerant phenotype of *Arabidopsis* transgenic plants produced by *GmCAMTA12* overexpression [22] and sensitive phenotypes by *GmCAMTA2* and *GmCAMTA8* overexpression (Figure 4) under drought stress conditions, *GmCAMTA2*, *GmCAMTA8*, and *GmCAMTA12* may be associated with drought tolerance response by different regulatory mechanisms. Our results suggest that GmCAMTA2 and GmCAMTA8 play an important role in both the development process and drought stress responses by circadian regulation.

## 4. Materials and Methods

### 4.1. Plant Materials and Growth Conditions

We used the soybean (*Glycine max* cv. Williams 82) and *Arabidopsis* (*Arabidopsis thaliana* Col-0 ecotype) plants in this study. The soybean plants were cultured in a growth chamber at 28 °C under long-day conditions (16 h light/8 h dark), and *Arabidopsis* plants in a growth chamber at 22 °C under long-day conditions (16 h light/8 h dark). To test the expression levels of *GmCAMTAs*, the two-week-old soybean plants were hydroponically cultured in the 1/2 MS medium with circadian rhythms. We used the two-week-old soybean plants in the 1/2 MS medium for various abiotic stress treatments. For salt stress treatment, the two-week-old soybean plants were transferred uniformly in the 1/2 MS liquid medium containing 150 mM NaCl (salt stress). For cold stress treatment, the two-week-old soybean plants in the 1/2 MS medium were placed on ice (approximately 2 °C). For ABA hormonal treatment, the two-week-old soybean plants were transferred uniformly in the 1/2 MS liquid medium. And then, 100 µM ABA was sprayed on the leaves of soybean plants. To treat soybean or Arabidopsis plants to drought stress, the two-week-old soybean or 10-day-old *Arabidopsis* plants, after removing the medium, were transferred to the Petri dish. And then, plants were placed in a growth chamber under long-day conditions (16 h light/8 h dark). Stress treatments were applied at ZT 3. After stress treatments, we performed the plant sampling at indicated different times. For the expression analysis of *GmCAMTAs* in different tissues, the soybean plants, after sowing seeds, were grown in the greenhouse under the seasonal cultivation period.

### 4.2. Identification of GmCAMTA Genes in Soybean

The *GmCAMTA* genes were identified from the soybean genome database at Phytozome version 13 (https://phytozome-next.jgi.doe.gov/, accessed on 1 July 2019). The specific primers for the qRT-PCR and RT-PCR analyses were designed with the IDT (Integrated DNA Technologies) service website (https://sg.idtdna.com/pages, accessed on 10 August 2020) based on the sequence information of *GmCAMTAs* (Appendix A). The full-length coding sequences of *GmCAMTA2* (3309 bp) and *GmCAMTA8* (2736 bp) were amplified by PCR using cDNA from soybean plants, and then cloned into the *pMLBart* binary vector under the control of the *CaMV 35S* promoter, which contained a BASTA resistance gene (Appendix A). We confirmed the full-length coding sequences through DNA sequencing analysis.

### 4.3. Generation of GmCMATA-Overexpressing Arabidopsis Transgenic Plants

The *pMLBart-GmCAMTA2* and *pMLBart-GmCAMTA8* plasmids were transformed into *Arabidopsis* plants using the *Agrobacterium*-mediated floral-dipping methods. The *GmCAMTA2*- and *GmCAMTA8*-overexpressing transgenic plants were selected on the soil by spraying with 0.2% BASTA herbicide. The expression levels of *GmCAMTA2* and *GmCAMTA8* in *Arabidopsis* transgenic plants were analyzed using reverse transcription PCR (RT-PCR) with gene-specific primers (Appendix A).

### 4.4. Physiological Assay of Drought Stress

To test the physiological phenotypes under drought stress conditions, two-week-old plants grown in soil with sufficient water were not watered for 13 days. After re-watering, the recovery of the drought-treated plants was monitored. The assays for drought sensitivity were thrice-replicated experiments using at least 12 plants for each line in each experiment. To measure the transpirational water loss, the shoots of 4-week-old plants in the soil were detached from the root. After being placed on Petri dishes, it was weighed immediately. The fresh weights of detached leaves were measured periodically at the indicated times, and the percentages of water loss were calculated. The water loss assays were replicated three times, using at least four plants for each line in each experiment.

### 4.5. Analysis of Gene Expression

The total RNA was isolated from each plant sample using the RNeasy Plant Mini Kit (Qiagen, Hilden, Germany), according to the manufacturer’s instructions. To remove any genomic DNA contaminants, purified total RNA was treated with DNaseI (Sigma-Aldrich, St. Louis, MO, USA). For the RT-PCR and quantitative RT-PCR (qRT-PCR) analyses, 1 µg of total RNA was used for cDNA synthesis using the RevertAid First Strand cDNA Synthesis Kit (Thermo Fisher Scientific, Waltham, MA, USA), in accordance with the manufacturer’s protocol. The qRT-PCR analysis was performed using the QuantiSpeed SYBR No-ROX Kit (PhileKorea, Seoul, Republic of Korea), and the relative gene expression levels were automatically calculated using the CFX384 real-time PCR detection system (Bio-Rad Laboratories, Hercules, CA, USA). The expression levels of soybean *GmTUBULIN* and *Arabidopsis AtTUBULIN2* were used as the internal control. The qRT-PCR experiments were performed in three independent replicates. The gene-specific primers used are listed in Appendix A. To represent the heatmap, qRT-PCR data for each *GmCAMTA* gene were normalized on the *GmTUBULIN,* and the expression heatmap was performed with the Microsoft Excel program.

## 5. Conclusions

In summary, we characterized the biological roles of 15 *GmCAMTA* genes in soybean (*Glycine max* cv. Williams 82). A total of 15 *GmCAMTAs* were significantly expressed in accordance with circadian rhythms in the daily clock. The expression levels of most *GmCAMTAs* were induced during various abiotic stresses, including salt, drought, and cold stress, except ABA. Interestingly, *GmCAMTA2*, *GmCAMTA8*, and *GmCAMTA12* were more highly expressed in stem tissues. The overexpression of *GmCAMTA2* and *GmCAMTA8* in *Arabidopsis* suppressed plant tolerance under drought-stress conditions. The expression levels of stress-responsive genes were dramatically reduced by overexpressing *GmCAMTA2* and *GmCAMTA8* in drought stress responses. These results suggest that GmCAMTA2 and GmCAMTA8 function in tissue-specific and plant stress responses through rhythmic regulation under drought stress conditions.

## Figures and Tables

**Figure 1 ijms-24-11477-f001:**
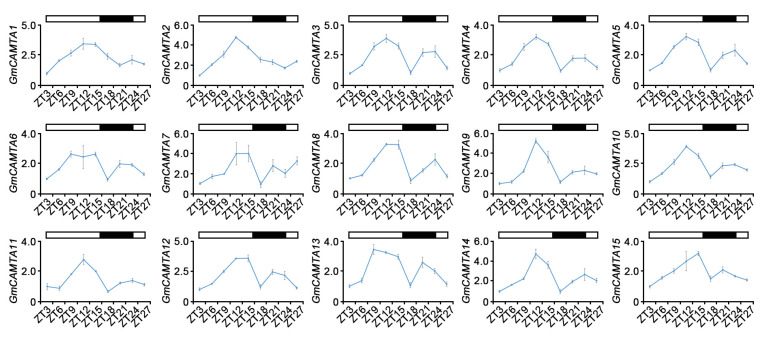
Transcriptional influence of *GmCAMTAs* in circadian rhythms. The expression levels of *GmCAMTAs* were analyzed for circadian regulation using qRT-PCR. The two-week-old seedlings of wild-type (WT) soybeans (*Glycine max* cv. Williams 82) were grown under long-day conditions (16 h light/8 h dark). Zeitgeber time (ZT) means a time of turning on a light. The white and black areas mark the approximate division of light and dark. The qRT-PCR analysis was performed with each *GmCAMTA*-specific primer (Appendix A). *GmTUBULIN* was used as an internal control for normalization. Error bars represent the SD of three biological replicates, each with three technical replicates.

**Figure 2 ijms-24-11477-f002:**
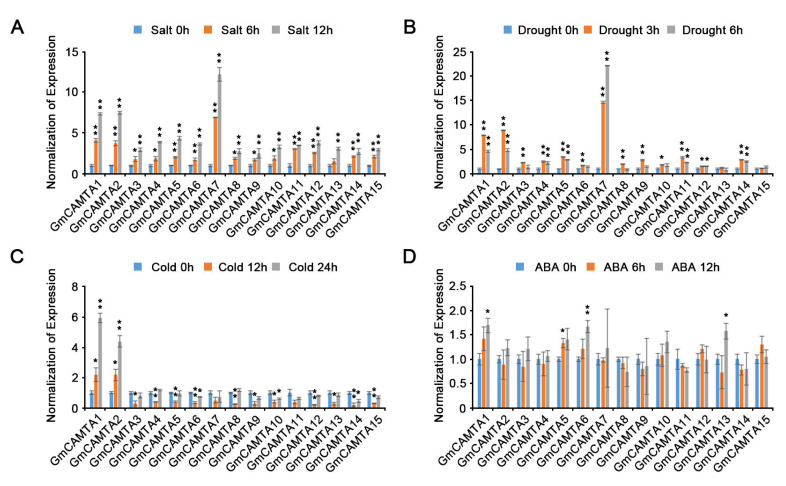
Effects of various abiotic stresses on the expression of *GmCAMTAs*. The expression levels of *GmCAMTAs* were analyzed in soybean seedlings during exposure to various abiotic stresses using qRT-PCR. The total RNA was extracted from two-week-old seedlings of WT soybeans under different stresses: (**A**) salt (150 mM NaCl), (**B**) drought, (**C**) cold (approximately 2 °C), and (**D**) 100 µM ABA. The qRT-PCR analysis was performed with each *GmCAMTA*-specific primer (Appendix A). *GmTUBULIN* was used as an internal control, and then the values of the expression levels were calculated by expression values of *GmCAMTAs* under non-stress conditions for normalization. Error bars represent the SD of three biological replicates, each with three technical replicates. Asterisks indicate significant differences compared with the expression of the zero indicated time point by Student’s *t*-test (* *p*-value  <  0.05, ** *p*-value  <  0.01).

**Figure 3 ijms-24-11477-f003:**
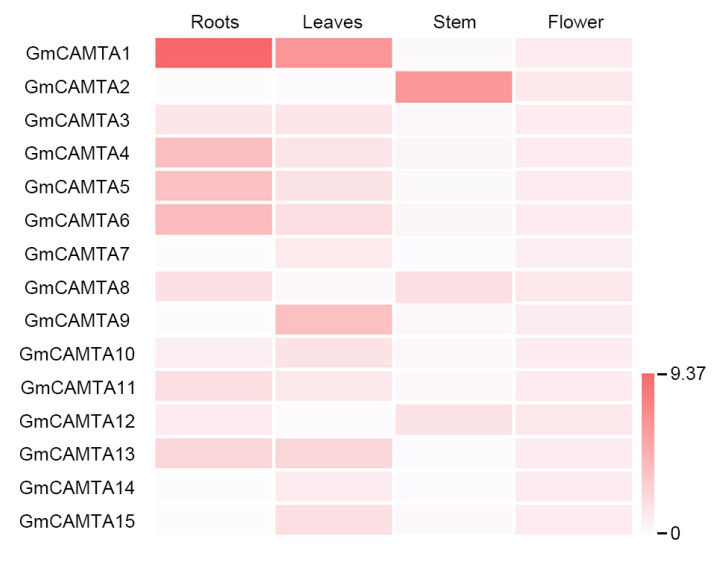
Heatmap analysis of *GmCAMTAs* expression in soybean tissues. The heatmap analysis based on qRT-PCR results showed the relative expression levels of *GmCAMTAs* in different tissues of soybean plants. The total RNA was extracted from various tissues, including roots, leaves, stem, and flower, at the V4 stages of WT soybeans grown under long-day conditions (16 h light/8 h dark). The qRT-PCR analysis was performed with each *GmCAMTA*-specific primer (Appendix A). *GmTUBULIN* was used as an internal control for normalization.

**Figure 4 ijms-24-11477-f004:**
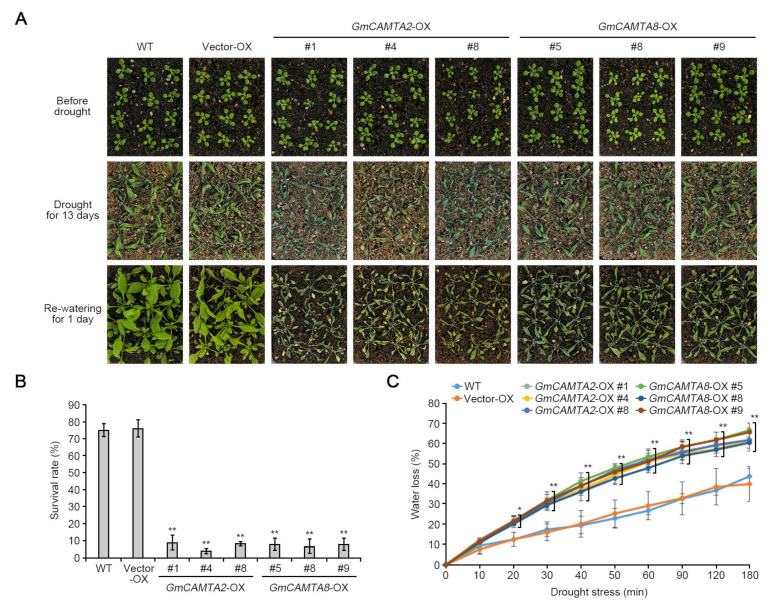
Effect of drought stress on *GmCAMTA2* and *GmCAMTA8* overexpressing transgenic *Arabidopsis*. (**A**) Phenotypes of *GmCAMTA2*-OX and *GmCAMTA8*-OX compared with WT and Vector-OX plants in response to drought stress and re-watering. The plants were grown in soil with sufficient water for two weeks (upper panels), then water was withheld for thirteen days (middle panels). The drought-stressed plants were then re-watered for one day (bottom panels). Each experiment comprised at least 12 plants, and three replicates were performed. (**B**) Comparison of survival rates (%) among *GmCAMTA2*-OX, *GmCAMTA8*-OX, Vector-OX, and WT plants against drought stress. Each experiment comprised at least 12 plants, and error bars represent the SD of three biological replicates. Asterisks indicate significant differences compared with WT plants by Student’s *t*-test (** *p*-value  <  0.01). (**C**) Analysis of water loss from detached leaves of three-week-old *GmCAMTA2*-OX, *GmCAMTA8*-OX, Vector-OX, and WT plants under drought stress conditions. Average values of water loss were measured at the indicated time points. Each experiment comprised at least four plants, and error bars represent the SD of three biological replicates. Asterisks indicate significant differences compared with WT plants by Student’s *t*-test (* *p*-value  <  0.05, ** *p*-value  <  0.01).

**Figure 5 ijms-24-11477-f005:**
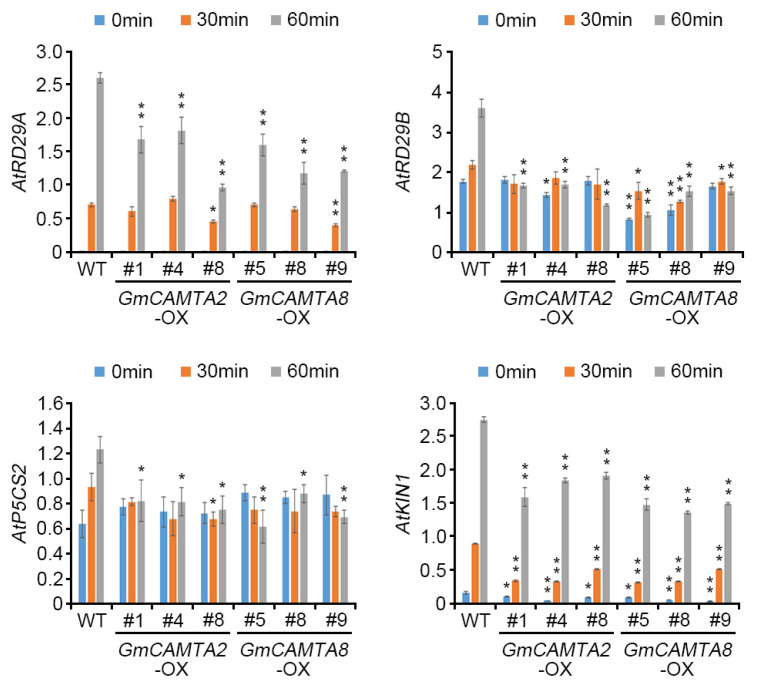
Effect on the transcriptions of *Arabidopsis* stress-responsive genes by the overexpression of *GmCAMTA2* and *GmCAMTA8* in drought stress responses. The expression levels of stress-responsive genes were analyzed using qRT-PCR in *GmCAMTA2*-OX and *GmCAMTA8*-OX plants under drought-stress conditions. Total RNA was extracted from 10-day-old *GmCAMTA2*-OX and *GmCAMTA8*-OX *Arabidopsis* at indicated time points to drought stress. The qRT-PCR analysis was performed with each gene-specific primer of the *Arabidopsis* stress-responsive genes, *AtRD29A*, *AtRD29B*, *AtP5CS2*, and *AtKIN1* (Appendix A). The expression of *Arabidopsis AtTUBULIN2* was used as an internal control for normalization. Error bars represent the SD of two biological replicates, each with three technical replicates. Asterisks indicate significant differences compared with expression levels in WT plants by Student’s *t*-test (* *p*-value  <  0.05, ** *p*-value  <  0.01).

## Data Availability

Data supporting the reported results can be made available upon request to the corresponding author.

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
