# Peer review of "Soybean Calmodulin-Binding Transcription Activators, GmCAMTA2 and GmCAMTA8, Coordinate the Circadian Regulation of Developmental Processes and Drought Stress Responses"

_ijms, 2023, doi:10.3390/ijms241411477_

Round 1

Reviewer 1 Report

The English language need to be checked and improved.

Author Response

[Reviewer 1]

This ARTICLE entitled "Soybean Calmodulin-binding Transcription Activators, GmCAMTAs, Negatively Coordinate the Circadian Regulation of Developmental Processes and Drought Stress Responses in Transgenic Arabidopsis" by Baek et al investigated the vast expression profiles of GmCAMTAs in circadian rhythms, abiotic stress and different tissues. The results provide a good foundation for understanding their functions. Additionally, GmCAMTA2 and GmCAMTA8 were functionally descripted by overexpressing in Arabidopsis. This manuscript will be interest to the researchers in this area and merit to be published after addressing some comments.

Major problems:

  1. The Introduction section should be improved, as many of the content have little to do with this study. However, the content was barely related to the Circadian rhythms.

[Response #1]

Thank you for your comments. As mentioned above, we deleted the unnecessary content for our study and added the circadian rhythms-related content in the introduction part of the revised manuscript.

  1. A heat map can be shown more directly when there is a lot of expressive analysis. In particular in Figure 3, it will help you compare the expression patterns of GmCAMTAs among different tissues.

[Response #2]

Thank you for your suggestions. As suggested by the reviewer, we have changed the bar graph to the heatmap in Figure 3. In addition, bar graph for gene expression in soybean tissues added to Supplementary Figure S2 in the revised supporting information.

  1. The nomenclature of GmCAMTAs showed be mentioned in the MM section, by according to the chromosomes or any other basises. Does GmACMTA12 was same as in references 31?

[Response #3]

Thank you for your comments. The nomenclature of GmCAMTAs in our studies is in the same notation according to the previously reported research (Wang et al., 2015). And GmCAMTA12 mentioned in both the previous report (Noman et al., 2019) and our studies are the same gene located in chromosome 17 of soybean.

References;

Wang, G.; Zeng, H.; Hu, X.; Zhu, Y.; Chen, Y.; Shen, C.; Wang, H.; Poovaiah, B.W.; Du, L. Identification and expression analyses of calmodulin-binding transcription activator genes in soybean. Plant soil. 2015, 386, 205-221. doi: 10.1007/s11104-014-2267-6.

Noman, M.; Jameel, A.; Qiang, W.D.; Ahmad, N.; Liu, W.C.; Wang, F.W.; Li, H.Y. Overexpression of GmCAMTA12 Enhanced Drought Tolerance in Arabidopsis and Soybean. Int J Mol Sci. 2019, 20, 4849. doi: 10.3390/ijms20194849.

  1. The author select GmCAMTA2 and GmCAMTA8 for functional analysis, why? Only based on their preferential expression in stem tissue is not sufficient. However, it was showed that GmCAMTA7 was more induced expression under drought stress in Figure 2B.

[Response #4]

Thank you for your comments. We have focused on the identification of the biological functions of soybean GmCAMTAs and are currently undergoing related research with GmCAMTA genes. In our initial study, we first took note of the high expression levels of GmCAMTA2, 8, and 12 in specific tissues, such as the stem. Since the previous report shows that GmCAMTA12 is involved in drought stress response (Wang et al., 2015), so we focused on the functional analysis of GmCAMTA2 and 8 in drought stress response.

As you indicated, our results showed that GmCAMTA7 is the most highly expressed in response to salt and drought stresses, and GmCAMTA1 and 2 genes are highly expressed in cold stress response (Figure 2). But we first cloned cDNAs for GmCAMTA1 and 2, so we started their functional studies first, such as the generation of Arabidopsis transgenic plants overexpressing GmCAMTA1 and 2 genes. We are currently studying whether GmCAMTA7 is associated with multiple signaling networks of various abiotic stresses by using ectopic expression and genome editing techniques.

References;

Wang, G.; Zeng, H.; Hu, X.; Zhu, Y.; Chen, Y.; Shen, C.; Wang, H.; Poovaiah, B.W.; Du, L. Identification and expression analyses of calmodulin-binding transcription activator genes in soybean. Plant soil. 2015, 386, 205-221. doi: 10.1007/s11104-014-2267-6.

  1. The hypersensitivity phenotypes of GmACMTA2-OX and GmACMTA8-OX lines were obvious, does the mutant lines exhibit an opposite phenotype?

[Response #5]

Thank you for your comments. We are currently conducting the generation of genome-editing soybean mutants using the CRISPR/Ca9 system to study the physiological and molecular functions of GmCAMTAs. However, currently, we do not have data for them yet. It would take a while to develop the stable genome-edited mutant lines.

Minor concerns:

  1. L22 “transcriptions” should be “transcripts” in the Abstract.

[Response #1]

Thank you for your suggestions. As suggested by the reviewer, we changed the “transcriptions” to “transcripts” in the revised manuscript.

  1. L139 the Latin Name should be addressed in the first time they appear.

[Response #2]

Thank you for your suggestions. As suggested by the reviewer, we removed the Latin name of the soybean species.

  1. Figure S2: The WT or Vector-OX lines were advised to be used as negative controls.

[Response #3]

Thank you for your suggestions. As suggested by the reviewer, we added RT-PCR data in the negative control (WT and Vector-OX) using specific primers of GmCAMTA2 and 8 genes in Supplementary Figure S3 of the revised manuscript.

  1. References: The format needs to be uniform, as more words are capitalized, Such as in Ref 3, Ref 4, Ref 5, Ref 10, ect.

[Response #4]

Thank you for your suggestions. As suggested by the reviewer, we modified the reference form in the revised manuscript according to the reference formatting of the IJMS journal.

Reviewer 2 Report

The present manuscript entitled “Soybean Calmodulin-binding Transcription Activators, GmCAMTAs, Negatively Coordinate the Circadian Regulation  of Developmental Processes and Drought Stress Responses in Transgenic Arabidopsiswritten well. Good concept, well organized experiment and nicely presented with photographs and diagrams. However, authors are advised to organize references according to MDPI style and improve discussion with supporting references of newly work carried out by researchers and published in high impact journals. In my opinion it will be better if authors also tested the function of these genes in soybean as well. If done then include if not then try to do in future study. English grammar also need to improve.

Minor English revision is required

Author Response

[Reviewer 2]

The present manuscript entitled “Soybean Calmodulin-binding Transcription Activators, GmCAMTAs, Negatively Coordinate the Circadian Regulation of Developmental Processes and Drought Stress Responses in Transgenic Arabidopsis” written well. Good concept, well organized experiment and nicely presented with photographs and diagrams. However, authors are advised to organize references according to MDPI style and improve discussion with supporting references of newly work carried out by researchers and published in high impact journals. In my opinion it will be better if authors also tested the function of these genes in soybean as well. If done then include if not then try to do in future study. English grammar also need to improve.

[Response ]

Thank you for your comments. As suggested by the reviewer, we modified the reference form in the revised manuscript according to the reference formatting of the IJMS journal.

We agree with the reviewer’s suggestion of functional analysis in soybean. We are currently conducting the generation of genome-editing soybean mutants using the CRISPR/Ca9 system to study the physiological and molecular functions of GmCAMTAs. However, currently, we do not have data for them yet. It would take a while to develop the stable genome-edited mutant lines.

Here, we edited English and attached an English editing certificate.

Reviewer 3 Report

The manuscript by Baek et al. considers the role of calmodulin-binding transcription activators in circadian rhythms and plant responses under different stress conditions. This work can be interesting for understanding of regulatory mechanisms in plants. However, I have some questions and remarks.

(1)    Figure 1 shows expression of GmCAMTAs for about 24 hours. This is too short period of time for investigation of circadian rhythms.

(2)    The circadian rhythms showed on Figure 1 only. Why the circadian rhythms were not investigated for other cases in Figures 2-5?

(3)    Why GmCAMTA2 and GmCAMTA8 were only investigated in experiments with drought (Figures 4 and 5)?

(4)    Why in Figure 2 3- and 6-hours water deficit called “drought”? The more detailed description of experiment should be added.

(5)    The description of experiment with could, salt and ABA treatment should be extended.

(6)    How experiment with “drought” on Petri dishes corresponds to a multi-day soil drought? Why variant of multi-day drought was not used for investigation of stress-responsive genes in Figure 5?

Author Response

[Reviewer 3]

The manuscript by Baek et al. considers the role of calmodulin-binding transcription activators in circadian rhythms and plant responses under different stress conditions. This work can be interesting for understanding of regulatory mechanisms in plants. However, I have some questions and remarks.

(1) Figure 1 shows expression of GmCAMTAs for about 24 hours. This is too short period of time for investigation of circadian rhythms.

[Response #1]

Thank you for your comments. We investigated the rhythmic expression of GmCAMTAs during 24 hours under long-day conditions, as Figure 1 shows. Although reviewers would think that 24 hours are too short a period of time for studying circadian rhythms response, recently many researchers reported that not only Arabidopsis genes (Mizoguchi et al., 2005) but crop genes, such as soybean (Lu et al., 2020; Su et al., 2022; Zhang et al., 2022), are involved in circadian rhythms-related responses via investigation for 24 hours. Our results are a sufficient basis for explaining the functions of GmCAMTAs in circadian rhythms.

References;

Mizoguchi, T.; Wright, L.; Fujiwara, S.; Cremer, F.; Lee, K.; Onouchi, H.; Mouradov, A.; Fowler, S.; Kamada, H.; Putterill, J.; Coupland, G. Distinct roles of GIGANTEA in promoting flowering and regulating circadian rhythms in Arabidopsis. Plant Cell. 2005, 17, 2255-2270. doi: 10.1105/tpc.105.033464.

Lu, S.; Dong, L.; Fang, C.; Liu, S.; Kong, L.; Cheng, Q.; Chen, L.; Su, T.; Nan, H.; Zhang, D.; Zhang, L.; Wang, Z.; Yang, Y.; Yu, D.; Liu, X.; Yang, Q.; Lin, X.; Tang, Y.; Zhao, X.; Yang, X.; Tian, C.; Xie, Q.; Li, X.; Yuan, X.; Tian, Z.; Liu, B.; Weller, J.L.; Kong, F. Stepwise selection on homeologous PRR genes controlling flowering and maturity during soybean domestication. Nat Genet. 2020, 52, 428-436. doi: 10.1038/s41588-020-0604-7.

Su, Q.; Chen, L.; Cai, Y.; Chen, Y.; Yuan, S.; Li, M.; Zhang, J.; Sun, S.; Han, T.; Hou, W. Functional Redundancy of FLOWERING LOCUS T 3b in Soybean Flowering Time Regulation. Int J Mol Sci. 2022, 23, 2497. doi: 10.3390/ijms23052497.

Zhang, Z.; Yang, S.; Wang, Q.; Yu, H.; Zhao, B.; Wu, T.; Tang, K.; Ma, J.; Yang, X.; Feng, X. Soybean GmHY2a encodes a phytochromobilin synthase that regulates internode length and flowering time. J Exp Bot. 2022, 73, 6646-6662. doi: 10.1093/jxb/erac318.

(2) The circadian rhythms showed on Figure 1 only. Why the circadian rhythms were not investigated for other cases in Figures 2-5?

[Response #2]

Thank you for your comments. In Figure 2, we measured the relative expression values of GmCAMTA genes under both natural and abiotic stress conditions, then we carried out the normalization of the relative expression values measured under abiotic stress conditions with the relative expression values at each time point measured under natural conditions. We modified the detailed explanation of Figure 2 in the revised manuscripts. To investigate the tissue specificity of GmCAMTA genes’ expression in Figure 3, we performed the gene transcript change in different tissues regardless of circadian rhythms. In Figure 4 and Figure 5, we investigated physiological analysis and functional characterization in drought stress response. Previously considerable research had been conducted into similar experimental conditions likewise our study (Nemchenko et al., 2006; Zhang et al., 2018; Baek et al., 2020; Ahn et al., 2021).

References;

Nemchenko, A.; Kunze, S.; Feussner, I.; Kolomiets, M. Duplicate maize 13-lipoxygenase genes are differentially regulated by circadian rhythm, cold stress, wounding, pathogen infection, and hormonal treatments. J Exp Bot. 2006, 57, 3767-3779. doi: 10.1093/jxb/erl137.

Zhang, D.; Wang, Y.; Shen, J.; Yin, J.; Li, D.; Gao, Y.; Xu, W.; Liang, J. OsRACK1A, encodes a circadian clock-regulated WD40 protein, negatively affect salt tolerance in rice. Rice (N Y). 2018, 11, 45. doi: 10.1186/s12284-018-0232-3.

Baek, D.; Kim, W.Y.; Cha, J.Y.; Park, H.J.; Shin, G.; Park, J.; Lim, C.J.; Chun, H.J.; Li, N.; Kim, D.H.; Lee, S.Y.; Pardo, J.M.; Kim, M.C.; Yun, D.J. The GIGANTEA-ENHANCED EM LEVEL Complex Enhances Drought Tolerance via Regulation of Abscisic Acid Synthesis. Plant Physiol. 2020, 184, 443-458. doi: 10.1104/pp.20.00779.

Ahn, H.R.; Kim, Y.J.; Lim, Y.J.; Duan, S.; Eom, S.H.; Jung, K.H. Key Genes in the Melatonin Biosynthesis Pathway with Circadian Rhythm Are Associated with Various Abiotic Stresses. Plants (Basel). 2021, 10, 129. doi: 10.3390/plants10010129.

(3) Why GmCAMTA2 and GmCAMTA8 were only investigated in experiments with drought (Figures 4 and 5)?

[Response #3]

Thank you for your comments. We have focused on the identification of the biological functions of soybean GmCAMTAs and are currently undergoing related research with GmCAMTA genes. In our initial study, we first took note of the high expression levels of GmCAMTA2, 8, and 12 in specific tissues, such as the stem. Since the previous report shows that GmCAMTA12 is involved in drought stress response (Wang et al., 2015), so we focused on the functional analysis of GmCAMTA2 and 8 in drought stress response.

References;

Wang, G.; Zeng, H.; Hu, X.; Zhu, Y.; Chen, Y.; Shen, C.; Wang, H.; Poovaiah, B.W.; Du, L. Identification and expression analyses of calmodulin-binding transcription activator genes in soybean. Plant soil. 2015, 386, 205-221. doi: 10.1007/s11104-014-2267-6.

(4) Why in Figure 2 3- and 6-hours water deficit called “drought”? The more detailed description of experiment should be added.

[Response #4]

Thank you for your comments. To investigate the gene expression in drought stress response, many researchers commonly called “drought” the stress when exposing plants to dry conditions (Freitas et al., 2019; Huque et al., 2021). As suggested by the reviewer, we described in detail the experimental process in the “Materials and Methods” part of the revised manuscript.

References;

Freitas, E.O.; Melo, B.P.; Lourenço-Tessutti, I.T.; Arraes, F.B.M.; Amorim, R.M.; Lisei-de-Sá, M.E.; Costa, J.A.; Leite, A.G.B.; Faheem, M.; Ferreira, M.A.; Morgante, C.V.; Fontes, E.P.B.; Grossi-de-Sa, M.F. Identification and characterization of the GmRD26 soybean promoter in response to abiotic stresses: potential tool for biotechnological application. BMC Biotechnol. 2019, 19, 79. doi: 10.1186/s12896-019-0561-3.

Huque, A.K.M.M.; So, W.; Noh, M.; You, M.K.; Shin, J.S. Overexpression of AtBBD1, Arabidopsis Bifunctional Nuclease, Confers Drought Tolerance by Enhancing the Expression of Regulatory Genes in ABA-Mediated Drought Stress Signaling. Int J Mol Sci. 2021, 22, 2936. doi: 10.3390/ijms22062936.

(5) The description of experiment with could, salt and ABA treatment should be extended.

[Response #5]

Thank you for your comments. As suggested by the reviewer, we described in detail the experimental process in the “Materials and Methods” part of the revised manuscript.

(6) How experiment with “drought” on Petri dishes corresponds to a multi-day soil drought? Why variant of multi-day drought was not used for investigation of stress-responsive genes in Figure 5?

[Response #6]

Thank you for your comments. Usually, the expression analysis of stress-responsive genes is investigated with the drought-treated samples for a short time using Petri dishes, laboratory bench, or growth chamber (Xiong et al., 2014; Noman et al., 2019). In addition, the physiological analysis of plant stress tolerance or sensitivity to drought stress was performed during a multi-day in soil conditions (Xiong et al., 2014; Noman et al., 2019). Thus, we used the general methods for expression analysis of stress-responsive genes in our study.

References;

Xiong, H.; Li, J.; Liu, P.; Duan, J.; Zhao, Y.; Guo, X.; Li, Y.; Zhang, H.; Ali, J.; Li, Z. Overexpression of OsMYB48-1, a novel MYB-related transcription factor, enhances drought and salinity tolerance in rice. PLoS One. 2014, 9, e92913. doi: 10.1371/journal.pone.0092913.

Noman, M.; Jameel, A.; Qiang, W.D.; Ahmad, N.; Liu, W.C.; Wang, F.W.; Li, H.Y. Overexpression of GmCAMTA12 Enhanced Drought Tolerance in Arabidopsis and Soybean. Int J Mol Sci. 2019, 20, 4849. doi: 10.3390/ijms20194849.

Reviewer 4 Report

The manuscript contain novel information regarding circadian regulation and organ specific gene expression of GmCAMTAs, however there are several flaws regarding methodology, the logic of the experiments  and interpretation of  the results. It needs substantial revisions before being considered for publication, as it will be compared with other high quality papers published on the subject in IJMS, besides some papers report increased resistance to drought in transgenic plants overexpressing CAMTAs whereas in this manuscript the authors claim that “ GmCAMTA2 and GmCAMTA8 act as negative regulators in drought stress  responses” but the evidence is not convincing enough .

The English should be substantially revised, as at places it is not clear what the authors are meaning to say, for example in the abstract “ To characterize the biological processes of soybean CAMTA (GmCAMTA) family members in response to abiotic stress, we identified 15 GmCAMTA genes from soybean…” (actually these genes were previously identified); introduction “The major function of the Ca2+ ion is to enhance the plant’s tolerance to stabilize cell walls and membranes against various plant stresses” ; “In many crops, soybean is highly sensitive to drought stress responses through  various morphological changes”; etc .

 My Major remarks are on the unclear description of the methods. Lines 397-399 “To test the  expression levels of GmCAMTAs, the two-week-old soybean plants were hydroponically cultured in the 1/2 MS medium with circadian rhythms” – when circadian rhytms began – after 2 weeks? At which hour of the daylight  was applied stress treatment? The authors establish circadian fluctuations in the expression of CAMTAs so this is very important. Drought stress treatment is completely unclear – “In the treatment of drought stress, the two-week-old soybean or 10-day-old Arabidopsis plants after removing the medium were placed in a growth chamber” ???  If growth and treatment is not explained in sufficient details, the results are doubtful. Moreover, not enough evidence is given concerning the physiological response to the applied stresses – “To measure the transpirational water loss, leaves from 4-week-old plants in  soil were detached and placed on Petri dishes”  - unconvincing. Please rewrite the MM section giving enough details.

Results: In this section, usually there are not references. The authors did not indicate from which tissues are the samples analyzed for circadian rhytms, as further they report some tissue specificity. These is a second peak in the expression of some CAMTAs left without comments. For stress treatments, the results had not considered the rhythmical changes of GmCAMTAs expression which may be over imposed on the stress response, and again – mRNA from which tissue? I miss the logic of the experiments – firstly should be the organ specificity, next – circadian rhytms which may differ comparing organs ( maybe stronger in leaves than in roots), next – the stress response with appropriate circadian controls. How the overexpressed genes were chosen, why exactly GmCAMTA2 and  8? How the treatment was chosen? It seems severe stress, however the effects may be quite different if moderate drought stress has been applied. Not enough physiological evidence about stress severity is given which is absolutely necessary if the results differ somehow from the expected ones.   

The English must be improved

Author Response

[Reviewer 4]

The manuscript contain novel information regarding circadian regulation and organ specific gene expression of GmCAMTAs, however there are several flaws regarding methodology, the logic of the experiments  and interpretation of  the results. It needs substantial revisions before being considered for publication, as it will be compared with other high quality papers published on the subject in IJMS, besides some papers report increased resistance to drought in transgenic plants overexpressing CAMTAs whereas in this manuscript the authors claim that “ GmCAMTA2 and GmCAMTA8 act as negative regulators in drought stress  responses” but the evidence is not convincing enough .

The English should be substantially revised, as at places it is not clear what the authors are meaning to say, for example in the abstract “ To characterize the biological processes of soybean CAMTA (GmCAMTA) family members in response to abiotic stress, we identified 15 GmCAMTA genes from soybean…” (actually these genes were previously identified); introduction “The major function of the Ca2+ ion is to enhance the plant’s tolerance to stabilize cell walls and membranes against various plant stresses” ; “In many crops, soybean is highly sensitive to drought stress responses through  various morphological changes”; etc .

[Response]

Thank you for your comments. As suggested by the reviewer, we edited some sentences that reviewers pointed out in the abstract and introduction parts.

 My Major remarks are on the unclear description of the methods. Lines 397-399 “To test the  expression levels of GmCAMTAs, the two-week-old soybean plants were hydroponically cultured in the 1/2 MS medium with circadian rhythms” – when circadian rhytms began – after 2 weeks?

[Response]

Thank you for your comments. Previous studies used the 2-week-old soybean seedlings to investigate the expression patterns of target genes in abiotic stress responses (He et al., 2019; Zhao et al., 2019). Accordingly, we investigated the circadian rhythmical expression of GmCAMTA genes in 2-week-old soybean seedlings.

References;

He, Q.; Cai, H.; Bai, M.; Zhang, M.; Chen, F.; Huang, Y.; Priyadarshani, S.V.G.N.; Chai, M.; Liu, L.; Liu, Y.; Chen, H.; Qin, Y. A Soybean bZIP Transcription Factor GmbZIP19 Confers Multiple Biotic and Abiotic Stress Responses in Plant. Int J Mol Sci. 2020, 21, 4701. doi: 10.3390/ijms21134701.

Zhao, M.J.; Yin, L.J.; Liu, Y.; Ma, J.; Zheng, J.C.; Lan, J.H.; Fu, J.D.; Chen, M.; Xu, Z.S.; Ma, Y.Z. The ABA-induced soybean ERF transcription factor gene GmERF75 plays a role in enhancing osmotic stress tolerance in Arabidopsis and soybean. BMC Plant Biol. 2019, 19, 506. doi: 10.1186/s12870-019-2066-6.

At which hour of the daylight was applied stress treatment? The authors establish circadian fluctuations in the expression of CAMTAs so this is very important.

[Response]

Thank you for your comments. Stress treatments were applied at ZT3. After stress treatments, we performed the sampling the seedlings at indicated different time points. We described in detail the experimental process in the “Materials and Methods” part of the revised manuscript.

Drought stress treatment is completely unclear – “In the treatment of drought stress, the two-week-old soybean or 10-day-old Arabidopsis plants after removing the medium were placed in a growth chamber” ???  If growth and treatment is not explained in sufficient details, the results are doubtful. Moreover, not enough evidence is given concerning the physiological response to the applied stresses – “To measure the transpirational water loss, leaves from 4-week-old plants in soil were detached and placed on Petri dishes”  - unconvincing. Please rewrite the MM section giving enough details.

[Response]

Thank you for your comments. As suggested by the reviewer, we described in detail the experimental process (Jiang et al., 2012) in the “Materials and Methods” part of the revised manuscript.

References;

Jiang, Y.; Liang, G.; Yu, D. Activated expression of WRKY57 confers drought tolerance in Arabidopsis. Mol Plant. 2012, 5, 1375-1388. doi: 10.1093/mp/sss080.

Results: In this section, usually there are not references. The authors did not indicate from which tissues are the samples analyzed for circadian rhytms, as further they report some tissue specificity. These is a second peak in the expression of some CAMTAs left without comments. For stress treatments, the results had not considered the rhythmical changes of GmCAMTAs expression which may be over imposed on the stress response, and again – mRNA from which tissue? I miss the logic of the experiments – firstly should be the organ specificity, next – circadian rhytms which may differ comparing organs ( maybe stronger in leaves than in roots), next – the stress response with appropriate circadian controls. How the overexpressed genes were chosen, why exactly GmCAMTA2 and  8? How the treatment was chosen? It seems severe stress, however the effects may be quite different if moderate drought stress has been applied. Not enough physiological evidence about stress severity is given which is absolutely necessary if the results differ somehow from the expected ones.  

[Response]

Thank you for your comments. We modified the explanation for Figure 1 and 2 in the results and the materials and methods parts of the revised manuscripts. To investigate the expressional change of GmCAMTA genes in circadian rhythms and stress responses, we used the 2-week-old whole seedling, including shoot and root (He et al., 2019; Zhao et al., 2019).

We have focused on the identification of the biological functions of soybean GmCAMTAs and are currently undergoing related research with GmCAMTA genes. In our initial study, we first took note of the high expression levels of GmCAMTA2, 8, and 12 in specific tissues, such as the stem. Since the previous report shows that GmCAMTA12 is involved in drought stress response (Wang et al., 2015), so we focused on the functional analysis of GmCAMTA2 and 8 in drought stress response.

References;

He, Q.; Cai, H.; Bai, M.; Zhang, M.; Chen, F.; Huang, Y.; Priyadarshani, S.V.G.N.; Chai, M.; Liu, L.; Liu, Y.; Chen, H.; Qin, Y. A Soybean bZIP Transcription Factor GmbZIP19 Confers Multiple Biotic and Abiotic Stress Responses in Plant. Int J Mol Sci. 2020, 21, 4701. doi: 10.3390/ijms21134701.

Zhao, M.J.; Yin, L.J.; Liu, Y.; Ma, J.; Zheng, J.C.; Lan, J.H.; Fu, J.D.; Chen, M.; Xu, Z.S.; Ma, Y.Z. The ABA-induced soybean ERF transcription factor gene GmERF75 plays a role in enhancing osmotic stress tolerance in Arabidopsis and soybean. BMC Plant Biol. 2019, 19, 506. doi: 10.1186/s12870-019-2066-6.

Wang, G.; Zeng, H.; Hu, X.; Zhu, Y.; Chen, Y.; Shen, C.; Wang, H.; Poovaiah, B.W.; Du, L. Identification and expression analyses of calmodulin-binding transcription activator genes in soybean. Plant soil. 2015, 386, 205-221. doi: 10.1007/s11104-014-2267-6.

Round 2

Reviewer 1 Report

All my concerns have been will responsed.

It's quite fine.

Author Response

Thank you for your review.

Reviewer 3 Report

Authors have improved the manuscript. I have no other questions or comments.

Author Response

Thank you for your review.

Reviewer 4 Report

The revised version has been substantially improved. Only minor remarks:

Abstract line 23 - "except ABA" - not clear, please rephrase

line 163 -"under natural and abiotic stress conditions" - better use optimal instead of natural as stress is also naturally encountered

line 209 - "that" should be omitted

line 379 - "ABA stress treatment" - ABA (hormonal) treatment 

see minor remarks

Author Response

The revised version has been substantially improved. Only minor remarks:

Abstract line 23 - "except ABA" - not clear, please rephrase

[Response]

Thank you for your comments. We described the "except ABA" in the abstract part because ABA did not influence the expression of most GmCAMTA genes in soybean. As suggested by the reviewer, we removed the "except ABA" in the revised manuscripts to prevent confusion.

line 163 -"under natural and abiotic stress conditions" - better use optimal instead of natural as stress is also naturally encountered

[Response]

Thank you for your comments. We changed from the "natural" to the “non-stress” in the revised manuscripts.

line 209 - "that" should be omitted

[Response]

Thank you for your comments. We removed the "that" in the revised manuscripts.

line 379 - "ABA stress treatment" - ABA (hormonal) treatment

[Response]

Thank you for your comments. We changed from the "ABA stress treatment" to the “ABA hormonal treatment” in the revised manuscripts.